# A Scoping Review of Psychosocial Risks to Health Workers during the Covid-19 Pandemic

**DOI:** 10.3390/ijerph18052453

**Published:** 2021-03-02

**Authors:** Paula Franklin, Anna Gkiouleka

**Affiliations:** 1Occupational Safety & Health and Working Conditions Unit, European Trade Union Institute (ETUI), Boulevard du Roi Albert II, 1210 Brussels, Belgium; 2Department of Public Health & Primary Care, University of Cambridge, Cambridge CB1 8RN, UK; ag2192@medschl.cam.ac.uk

**Keywords:** healthcare workers, occupational health, psychosocial risks, scoping review, Covid-19

## Abstract

The Covid-19 pandemic has exposed health workers to a diverse set of hazards impacting their physical, psychological and social wellbeing. This review aims to provide an overview of the categories of the psychosocial risk factors and hazards affecting HCWs during the Covid-19 pandemic and the recommendations for prevention. We used the scoping review methodology to collate categories of psychosocial risks, the related health outcomes, interventions, and data gaps. The review was conducted on global peer-reviewed academic and authoritative grey literature, published between 1. January–26. October 2020; in total, 220 articles were included into the review and the subsequent analysis. Analysis of the extracted data found PSRs related to four sources: personal protective equipment (PPE), job content, work organisation, and social context. is. Women health workers and nurses reported worst health outcomes. Majority of the research to date concerns health workers in secondary care, while data on psychosocial risks at primary and community-based settings are scarce. However, the emerging research implies that the pandemic creates psychosocial risks also to non-clinical health workers. The intervention and mitigation measures address individual and organisational levels. Preventative and mitigating measures for social and societal risks—such as staff shortages, intersecting inequalities, and financial stressors require further research.

## 1. Introduction

It has been a year since the World Health Organization (WHO) declared the novel coronavirus (Covid-19) outbreak a pandemic [1], and we have reached a point where we can assess the impact of the pandemic on healthcare systems since there is a large amount of empirical evidence documenting the challenges and the disruptions. The findings on the effects of the pandemic on health care workers (HCWs) highlight the multiple ways in which the Covid-19 pandemic poses a significant threat to their physical, psychological and social well-being [2,3].

This scoping review engages with this evidence and offers an overview of categories of psychosocial risks affecting medical and non-medical healthcare workers, the reported health outcomes associated with the risks, and the intersecting inequalities in negative health outcomes among the workers. Moreover, the review offers timely conclusions regarding the ways that organisations can support healthcare workers and ameliorate their physical and psychological burden, as well as signposts emerging issues for further research. Reviewing the factors impacting healthcare workers’ psychological wellbeing can support the development of measures to prevent and mitigate psychosocial risks, and to create healthy working environments [4,5]. In addition, the research on effects of Covid-19 continues to rapidly evolve, and therefore updated reviews are needed [6]. In this context, the primary aim of this study is to categorise data and evidence on PSR to health workers, and the related prevention measures. The secondary aim is to identify knowledge gaps.

Psychosocial risks (PSR) refer to the design and management of work, and its social and organizational context that have the potential to cause harm to workers [7]. Exposure to PSR can affect a worker’s psychological and physical health through a stress-mediated pathway. In addition, the health and resiliency of an organisation (e.g., absenteeism, high turnover, and organisational commitment) can be affected [8]. The sources of psychosocial risks are numerous, including:(1)*Job content*, e.g., conflicting demands, lack of role clarity, lack of training and development opportunities, and lack of workers’ influence over the way the job is done.(2)*Work organisation and management*, e.g., excessive workloads and work intensity, lack of workers’ involvement in making decisions that affect the worker (autonomy), poorly managed organisational changes, ineffective communication, working time arrangements, and poor work-life balance.(3)*The social context of the job*, e.g., lack of support from management or colleagues, psychological and sexual harassment, third-party violence, and job insecurity [9].

Health workers are known to be exposed to a variety of PSR at work [10,11,12,13,14,15], and communicable disease outbreaks exacerbate the risks [16,17]. Nurses as an occupational category, are found to be particularly exposed to PSR [18,19,20]. Exposure to physical hazards and psychosocial risks in healthcare arises from working overtime, work overload, and time pressure [21,22,23], an insufficient number of rest breaks and days away from work, leading to poor work-life balance [14,24], shiftwork [25,26], low wages and job insecurity [27,28,29], and exposure to adverse social behaviour, such as violence and harassment [30,31]. Precarious employment is increasing, particularly in the elderly- and long-term care sector with many migrant workers [32,33]; this type of employment is a source of financial stress due to income insecurity which can affect the health of workers [34]. A rapid review and meta-analysis of major research studies into the psychological effects on clinicians working in past outbreaks (severe acute respiratory syndrome SARS; Middle East respiratory syndrome MERS, Ebola virus disease; influenza A virus subtype H1N1; influenza A virus subtype H7N9) [4] found that risk factors for psychological distress included being younger, being more junior, being the parents of dependent children, or having an infected family member. Longer quarantine, lack of practical support, and stigma also contributed.

Measures addressing different aspects of work environment combined with individual interventions are shown to be the most effective solution to prevent PSR; single measures, especially when mainly targeting the individual worker, do not prove very effective [35,36]. Specific preventative measures for PSR include provision of training to workers, changes in work organization, redesigning of work areas, confidential counselling, changes to working time arrangements, and conflict resolution procedures [37]. The measures that successfully managed the effects of the psychological effects on clinicians working in past outbreaks included: clear communication, access to adequate personal protection, adequate rest, and both practical and psychological support [4]. Worker participation in the planning and management of changes in work organization, including occupational safety and health (OSH), have been linked to improvements in job satisfaction which in turn has a strong association with workers’ health. This is due to increased job control, the awareness about the work organization and understanding about work roles and processes, and an involvement in mitigating workplace challenges [14,37,38].

In the following we first, provide information on the recorded negative health outcomes to HCWs during the pandemic; second, highlight how PPE can be a source of PSR; third, describe specific factors within three sources of PSR - job contents, work organisation, and social context; and fourth, discuss the preventative and mitigating measures for PSR that are identified in the studies, as well as areas for further research.

## 2. Materials and Methods

### 2.1. Scoping Review Design and Data Collection

The scoping review method is used to describe existing literature and other sources of information, and commonly include findings from a range of different study designs and methods. The method can be a particularly useful approach when the information on a topic has not been comprehensively reviewed, which is the case with the emerging evidence regarding PSR in healthcare during the Covid-19 pandemic [39]. This scoping review employed the recommended five-steps framework for scoping reviews as introduced by Arksey and O’ Malley in 2005 [40] and further developed in recent work [41,42,43]. As per the framework, the steps included:(1)Identifying the research question with a broad scope: What are the main categories and factors of psychosocial risks to healthcare workers during the Covid-19 pandemic and the related prevention measures?(2)Identifying relevant studies.(3)Selecting studies as per the study protocol.(4)Charting the data. Relevant information was extracted into an Excel sheet from the reviewed literature.(5)Collating, summarising, and reporting the results in tables and charts according to key themes, with an analytical summary of the findings.

### 2.2. Identification of Relevant Studies

The review protocol defined ‘healthcare workers’ (HCWs) broadly, and included health service providers, such as doctors, nurses, midwives, public health professionals, medical and non-medical technicians, personal care workers, and community health workers. It also included health management and support workers in health systems, such as cleaners, drivers, hospital administrators, district health managers and social workers, and other occupational groups in health-related activities. Studies on acute and long-term care facilities were included, as was community-based care, social care and home care [44].

Our searches were conducted in the PubMed electronic database and via the Google Scholar search engine to ensure both biomedical research as well as non-biomedical research and grey literature were located. Given that the two systems do not search identical data [45], we considered that their combined use would lead to less overlaps as well as access to conference proceedings, pre-print archives and institutional repositories. We were interested in that literature beyond academic journals because it contains guidelines, health reports on interventions and policy recommendations which are often produced by non-academic research bodies (e.g., non-governmental organisations or professional associations). Therefore, we used a combined search of key words reading the emerging literature and Medical Subject Headings (MeSH). The following keywords were used as search terms: for context “Covid-19” (MeSH), for the professional categories “healthcare staff” or “healthcare worker” (identified in the literature) or “community health workers” (MeSH), and for the domains “psychosocial” (MeSH) or “occupational health” (MeSH). The keywords were used to search across all fields in articles that were published in English, due to it being the common language between the researchers, and from 1 January 2020 due to research time limitations. The search took place between 16 and 26 October, resulted in 1111 papers and included peer-reviewed academic articles, reports and grey literature from prominent public health actors (e.g., WHO). Given the diversity of the publication types and the use of commentaries and opinion papers, we considered that assessing the methodological quality of the papers was not feasible in a harmonised and meaningful way. This is a common feature of scoping reviews and makes them different to systematic reviews [41] We proceeded with manual searches of the reference lists of the identified reviews to cover the breadth of the existing evidence. The manual search led to the addition of another 24 articles published within that time and fulfilling the study protocol to the full-text reading.

### 2.3. Study Selection

The criteria used for the selection of studies were broad: articles were included for full-text reading if they focused on the psychosocial factors affecting HCWs; or on the mental health and psychosocial well-being of HCWs; or on relevant interventions; or on overall recommendations for the protection of healthcare staff that included psychosocial well-being. Articles were excluded if they focused solely on the protection of healthcare staff from infection; screening and monitoring processes; or on recommendations regarding the ultimate provision of patient care. Articles on psychosocial risks to HCWs during other infectious disease outbreaks (e.g., SARS, MERS, Ebola) were excluded; as were studies focusing solely on individual HCW resilience. 

After the first screening process of titles and abstracts, out of the 1134 identified articles, 290 were included for full-text reading. The reading resulted in 220 articles that met the criteria for the extraction of relevant information (see Figure 1). The omissions and additions of articles were discussed and agreed between the researchers. The reasons for omitting articles included: they were not addressing psychosocial risks, not studying Covid-19 context, studies focusing on patients or general populations and not on HCWs, self-care handbook, and full text not available in English.

This diagram describes the process through which articles were identified for and excluded from the study.

### 2.4. Data Chart and Results Overview

The aim of a scoping review is the presentation of an overview of the reviewed literature [40]. In our case, the main focus was an overview of the categories of the psychosocial risk factors and hazards affecting HCWs during the Covid-19 pandemic and the recommendations for prevention. To achieve this aim, we created a data chart template in Excel, that included for each article: the title, authors, online location (web link), the source of publication, the type of the study/publication, the abstract, the country, the region and the setting where the study took place, the occupations of the study participants, the sample characteristics, health outcomes, the main findings, conclusions and recommendations. The extraction and categorisation of the data for each article was conducted by one of the reviewers, while the second reviewer assessed the charting process to identify potential errors or to resolve any conflicts. Finally, the extracted data were used to produce a summary overview [42] according to the key sources and themes [40] presented in the results section; health outcomes and main sources of PSR based on the main findings; and preventative measures based on the conclusions and recommendations that are discussed in the studies.

### 2.5. Study/Publication Types

There is a variety of study designs and publication types included in the selected articles; studies with primary data and commentary essays, reviews and opinion pieces. Specifically, there are 131 studies with primary data. The majority of them are cross-sectional studies (98), case studies (intervention evaluations) (15), and qualitative studies (14). There are also 31 review papers (literature, scoping and systematic), 56 articles without primary data but with peer-reviewed references (opinion articles, editorials, commentary papers), and two non-academic papers (grey literature).

### 2.6. Regions and Countries

The list of publications includes 28 articles focusing on countries in the European Union, among which the majority (13 articles) focus on Italy, six on Spain, and three on France. The rest of studied countries are Finland (one), Germany (one), Greece (one), Ireland (one), Poland (one), and Portugal (one).

Another 136 publications refer to countries outside the European Union with the majority focusing on China (including Hong Kong and Taiwan) (35), USA (19), UK (16), India (seven), Pakistan (six), Turkey (six), Canada (three), Singapore (six), Iran (four), Australia (four), Saudi Arabia (four), South Africa (three), Israel (two), Malaysia (two), Philippines (two), Bangladesh (one), Brazil (one), Egypt (one), Japan (one), Lebanon (one), Mexico (one) Nigeria (one), Oman (one), Palestine (one), Yemen (one ) and Nepal (one). One publication focuses on the broader region of Americas and another one on Africa while there are four publications focusing on multiple countries outside the EU. Finally, there are three studies on multiple countries both in and outside the EU, and another three with a global scope. Fifty of the included publications do not have an exclusive geographical focus as they are commentaries, reviews, or opinion pieces of general interest.

### 2.7. Studied Settings, Occupations and Sample Characteristics

The majority of the included studies with primary data focus on hospitals and medical centres (101), including cancer centres, paediatric hospitals, dental hospitals, and cardiac centres. There are only two articles with an exclusive focus on primary care settings and seven focusing on community settings (e.g., Alzheimer’s care homes, community pharmacies). Regarding the studied occupations, 63 studies focus on the broad category of medical HCWs, 29 on nurses, and 11 on doctors, while 30 studies examine medical and non-medical HCWs. More information on study settings and occupations is presented in Table 1. In the included studies, women are usually over-represented within the samples, and people who are married or living with partners. Level of education ranges according to the examined professions, however, graduates and post-graduates are usually not the majority. Regarding years of experience, in most studies the HCW have more than five years of service.

## 3. Results

### 3.1. Negative Outcomes for Healthcare Workers

The Covid-19 pandemic and the disruptions it has caused in the social and working life of HCWs has had a significant impact on their overall well-being (see Figure 2). Evidence from 55 empirical papers and systematic reviews suggests that higher prevalence rates of anxiety, stress and depression are observed among HCWs. Ten of the analysed publications focused on psychological trauma and post-traumatic stress symptoms. A study from China [46] one month after the outbreak showed that 3.8% of the studied sample reported PTSS, while other studies reported PTSS prevalence that reached 43% [47], 49% [48] and even 56.6% [49]. Commentaries and opinion articles stress that HCWs working at overtaxed healthcare systems are at elevated risk to PTSS and also that the situation is likely to deteriorate in the future after the second and third waves of the pandemic [50,51]. Sleep disturbances and insomnia are reported in 19 studies as a negative outcome of the pandemic that further affects HCWs’ well-being. Burnout, fatigue, physical and emotional exhaustion were reported in 14 studies.

While frontline HCWs are most impacted by the psychosocial risks and related negative health outcomes, the body of evidence reviewed shows that the mental health of all workers in the whole healthcare system can be impacted due to new policies and procedures and the risk of infection, including nurses and doctors in their usual hospital wards [52], ophthalmologists [53] medical imaging professionals [54], and nonmedical personnel, such as allied health professionals, pharmacists, technicians, administrators, clerical staff, and maintenance workers [55,56], as well as community-based nurses [57].

Women and nurses are usually the majority within the samples used in the analysed empirical studies. It emerges that women more often than not report higher levels of mental health problems including anxiety, stress, post-traumatic symptoms and depression compared to their men colleagues [46,58]. A study in Italy showed that women doctors are also more likely to experience compassion fatigue and burnout [59], and similarly in Turkey, a study in hospital workers showed that the risk of development of anxiety among women was 16.6 times higher than among men [60]. Similar findings are reported in the UK as well [61]. A cross-sectional survey of physician trainees in the US [62] found that that trainees exposed to COVID-19 patients were significantly more stressed and experienced greater burnout, and women trainees were more likely to have higher stress than men.

When HCWs evaluate themselves, nurses’ hopelessness and anxiety levels are higher than doctors’ [63] while they also report worse sleep quality [64]. A review and meta-analysis of 62 studies has showed that being a nurse is a risk factor for anxiety and depression [2]. Additional empirical evidence from Portugal shows that nurses report higher depression, anxiety and stress levels than the general population [65], while evidence from Germany suggests that nurses working in the Covid-19 wards report higher levels of stress, exhaustion, and depressive mood, as well as lower levels of work-related fulfilment compared to their colleagues in the regular wards [66]. A study among nurses in community-settings in Philippines revealed that community nurses experience of fear of Covid-19 is similar to nurses working in a hospital setting, with women appearing to be more fearful than men. With increased fear of Covid-19, the nurses’ psychological distress, as well as their organizational and professional turnover intentions increase [57].

A study among 376 nurses and physicians in Italy revealed a significant main effect of gender and occupational role on experienced psychosomatic and burnout symptoms, with males experiencing symptoms less frequently than females and physicians less frequently than nurses [16]. Additionally, a cross-sectional study among more than 4000 nurses in China revealed that female gender was a risk factor for higher levels of somatic symptoms [67]. Studies in Latin America show that female nurses are more often victims of stigmatization and physical assaults because they are more exposed to their communities (e.g., by using the public transport) [68]. A survey of medical staff involved in the 2019 Novel Coronavirus Disease Outbreak in China found that insomnia was higher in women and the low educated [69].

Nurses who work casually, make up a sizable proportion of many nations’ nursing workforce, and these staff are a vital workforce capacity to healthcare systems during pandemics [70]. Temporary, or travel/contract nurses without a core group of in-person colleagues, may feel less prepared for the challenges of the public health emergency [71], and have lower risk awareness, while having a higher risk of burnout and feelings of isolation [72].

### 3.2. Psychosocial Risks and Personal Protective Equipment (PPE)

The studies highlight how the fear of infection and transmission of the virus to family members, colleagues, and friends is the dominant worry among HCWs and the most important risk factor for their physical but also psychological well-being and mental health. According to the examined studies, there are three categories of risk factors that associate with PPE: anxiety related with the risk of infection due to the lack of PPE; anxiety and physical discomfort related to use routines and protocols; and practice disruptions and moral dilemmas.

#### 3.2.1. Anxiety Related with the Risk of Infection Due to Lack of Adequate PPE

Scarcity of or inadequate protective equipment for HCWs is systematically mentioned as a psychosocial risk factor triggering fears of infection and transmission of the virus. Commentary and opinion articles highlight how the lack of PPE leads in prolonged use or reuse of the same equipment intensifying feelings of insecurity and vulnerability among HCWs. A study in Italy [73] showed that although most respondents (77%) confirmed that PPE were readily available at the workplace, only 22% considered PPE adequate for quality and quantity; and PPEs were more readily available in high-risk specialty sectors but less so for HCWs with recent onset of symptoms. In Spain, a study among 157 hospital HCWs showed that 85.4% of the respondents stated that the lack of PPE generated an increase in stress and anxiety [74]. Similarly, a Canadian study among medical oncologists revealed that although the majority of the respondents expressed moderate-to-extreme concern about personally contracting Covid-19 and about family members or patients (or both) contracting covid-19 from them, 33% of them indicated no routine PPE use at their institutions and 69% indicated uncertainty about access to adequate PPE. Moreover, the availability of adequate PPE was reported as a factor reducing stress by 54% of a sample of 318 Palestinian HCWs [75].

#### 3.2.2. PPE Use Routines and Protocols Related with Anxiety and Physical Discomfort

The majority of the studies highlight that even when PPE is available, the ultimate and effective use of it is a source of anxiety for the HCWs. Many opinion articles highlight the need for training for proper donning/doffing of PPE as well as how these processes are a source of anxiety among HCWs [3,76,77]. A commentary paper on Pakistan stresses how the anxiety related with the use of PPE often prohibits drinking water and using the toilet [78]. Moreover, the prolonged use of PPE has been associated with significant physical discomfort like skin irritations [79], hypoxia and hypoglycemia [78]. Adherence observations of HCWs in a university hospital in Germany, revealed deficiencies in the use of recommended personal protective equipment (PPE) among HCW caring in Covid-19 wards during the first period of the SARS-CoV-2 pandemic. The deficits indicated a clear need for training in the correct use of PPE [80].

#### 3.2.3. PPE and Practice Disruptions

The prolonged use of PPE in different settings in healthcare is associated with practice disruptions, barriers in communication among HCWs and between HCWs and patients, conflicts among colleagues regarding PPE distribution and proper use, as well as difficulties in movement, and a constant need to adjust to changing protocols and safety guidelines [79,81]. Moreover, the use of PPE has been associated with moral dilemmas, for example, having to care of a suspected Covid-19 patient while not having appropriate PPE; or being unable to offer emotional support through facial expressions and touch [82,83]. For example, in speech therapy practice, one of the greatest challenges is the difficulty in communicating with patients through expressions and gestures to demonstrate movements, which is impaired due to the use of PPE. In addition, speech therapists and HCWs who provide swallowing therapy, are occupational groups at a high risk of infection, as the procedures performed require proximity to the patients’ faces and contact with the oral mucosa and body fluids, such as saliva and respiratory droplets [84,85]. All these factors are considered to increase anxiety and moral distress among HCWs and further affect their psychological well-being and mental health.

### 3.3. Psychosocial Risks and Job Content

The review found two categories of risk factors related to job content: redeployment and moral injury.

#### 3.3.1. Redeployment of HCWs to Covid-19 Care

The high number of patients during the pandemic has in many places resulted in the redeployment of HCWs to Covid-19 care wards. Redeployment in front-line settings and the need for adjustment, creates anxiety due to fear of infection [86,87,88,89]; and psychological stress due to sense of inadequacy in the new role [90]. The protocols of safety and infection control that involve isolation of the patients from their loved ones render HCWs as mediators, which further increases the demands on the workers [91]. Redeployment highlights how lack of influence over one’s job, and lack of role clarity create PSRs during the pandemic.

#### 3.3.2. Moral Injury

Managing scarce resources and having to take difficult decisions (e.g., triage, provision of ventilators for certain patients and not for others) create conflict between professional and ethical values on the one hand and HCW’s own safety or the availability of resources on the other [83,92,93]. Often, HCWs were aware that the decisions they need to take and the practices they follow because of the pandemic are against what is appropriate for the patient [93]. HCWs are exposed to continuous and increasing levels of emotional strain and moral injury [94]. Dealing with the Covid-19 patients’ feelings of distress, as well as with the loss of big numbers of patients and colleagues are identified as significant psychosocial risk factors affecting HCWs’ mental health [88,94,95]. A commentary [93] on nurses’ moral injury during the pandemic suggests that the shift from patient-centred ethics in healthcare to public health-centred ethics imposed by the current circumstances, represents a major challenge for nurses and triggers moral dilemmas in instances when practices of triage or patient prioritization take place and are in conflict with the duty to care for each particular patient.

What should be noted is that the emotional strain and the resulting stress and moral injury affect HCWs beyond those who work at the front-line with Covid-19 patients. In maternity services, for example, healthcare staff need to adjust in changed practices that may be in direct conflict with evidence, professional recommendations or moral beliefs and values of HCWs [96]. Studies also show that surgeons [91], dental health staff [97,98,99] mental health staff and trainees [100], ophthalmologists [53], and community pharmacists [101] are exposed to increased emotional strain and vulnerable to mora l injury.

### 3.4. Psychosocial Risks and Work Organisation

Work organisation refers to how work is planned, organised and managed, as well as to choices on a range of aspects such as work processes, job design, responsibilities, task allocation, work scheduling, work pace, rules and procedures, and decision-making processes [102]. The review found many stressors related to work organisation, including vague information, and changing safety protocols [81], lack of facilities for rest and taking care of personal hygiene [103], disruption to work routines, and isolation and quarantine practices for workers [90,104]. In addition, two inter-related aspects of PSR related to work organisation are work overload and lack of work-life balance.

#### Work Overload and Lack of Work-Life Balance

Work overload during the pandemic is a clear PSR in terms of patient numbers and hours; additional and unintended shifts reduce the autonomy of HCWs to decide their time use [105]. Staff shortages exacerbate the risk [57,106]. The proximal stressors, such as lack of childcare and poor work-family balance are a significant source of anxiety for HCWs [107,108]. Among the psychosocial risk factors that seem to be specific for women nurses is the uncertainty that their organization will support/take care of their personal and family needs if they develop infection, access to child-care during increased work hours and school closures, and support for other personal and family needs as work hours and demands increase (food, hydration, lodging, transportation) [107]. School closures have also resulted in teleworking, and some HCWs (e.g., allergists/immunologists) supervise clinical activities with children at home, which is challenging and requires new workflows for urgent and routine messages [109].

### 3.5. Societal and Social Demands as Sources of Psychosocial Risks

The examined studies reveal that HCWs are exposed to significant psychosocial pressure posed by the public; both their immediate social context (e.g., family, friends, neighbourhood, colleagues) and the broader societal context via media.

#### 3.5.1. Stigmatisation and Violence against HCWs

HCWs are in certain cases stigmatised in their communities as virus carriers. Studies in India and South Asia show that HCWs report fear of stigma or discrimination in their neighbourhood and are often afraid to go home after work [100] while also in Low- and Middle-Income Countries stigma leads to even eviction from accommodation and physical assaults [110]. A qualitative study among 18 nurses and doctors in Iran found that the reaction of the society was one of the HCWs’ main concerns together with stigmatising news in the media [105]. Similarly, HCWs and especially women and nurses have been also the victims of attacks and violence in their communities in Latin America [69]. Stigmatization as a psychosocial risk factor is also highlighted in commentaries and opinion papers focusing on the UK and the USA [82,85]. Moreover, studies show that HCWs are often the targets of intersectional processes of stigmatization across their professional roles, ethnicity, gender, and race. With nurses, women, Black, Asian, and minority ethnic (BAME) HCWs and all those combining these characteristics being more severely affected [69,94]. From a different viewpoint, HCWs often struggle with emotional distress that is caused by the representation of HCWs as heroes by the media and in the public discourse. It seems that these representations put pressure on HCWs to fulfil an ideal adding an extra level of anxiety [80] and also often associates with feelings of guilt for HCWs who do not offer services at the front-line against Covid-19 [110]

#### 3.5.2. Financial Stress

The reviewed articles report stress related to HCWs’ financial situation due to loss of income, possible salary cuts and furlough, and lay-offs [72,111,112]. In addition, many HCWs are the sole support of others in the family who may have lost employment, and under financial pressure, nurses may feel compelled to work more hours than is healthy [112].

This figure describes the sources of psychosocial risks, the categories, and the health outcomes.

### 3.6. Interventions and Recommendations

Interventions that seem to mitigate psychosocial risks involve management engagement and multi-dimensional approaches. Practical support is deemed necessary, such as space for rest and relaxation in the hospital, accommodation solutions for HCW who cannot stay at their homes for safety reasons, free meals, free parking services, and childcare. These measures should be coupled with psychological interventions, such as psychological support inside the healthcare settings, peer support groups, and psychological hotlines.

Experts highlight that models used for group support and reflection should be adjusted to the healthcare context and to the increased level of emotional stress caused by the pandemic. Specifically, buddying practices (i.e., one to one peer-support) need to be closely monitored, so that the same people are not overburdened. Moreover, managerial support and resourcing should be provided so that buddying is not seen or used as a substitute to adequate psychological or other support. Similarly, ‘Schwartz Rounds’ that provide an opportunity for staff from all disciplines across a healthcare organisation to reflect on the emotional aspects of their work, may not be the “right” solution at the peak of the pandemic, and should be adjusted in order to involve rounds in a virtual format to be run in smaller existing teams, and not across the whole organisations [100].

Recommendations for workers in Emergency Departments include among others clear, consistent and regular leadership and communication; staff safety, in terms of virus exposure; safe rest areas; rostering (1 week annual leave per 4–5 weeks through the peak of the pandemic has been recommended to ensure optimal recovery time for staff, and for them to maintain their capacity to fulfil their role); huddles in the beginning and debriefing at the end of a shift; training and education; peer supporters; and well-being drop in sessions [113].

Testing of HCWs for SARS-CoV-2 virus and the antibodies can reduce anxiety and stress [99,114].

Organising childcare was recommended as a form of support for HCWs [106,115], and work-life balance was reported to decrease the extent of the psychological consequences of the pandemic on HCWs [116,117].

A top-down workplace culture that enables bullying in response to HCWs’ concerns about safety at work culminates into a loss of trust in leadership, while consultation and engagement with HCWs [118], inclusive leadership [119], organisational support [120] and ‘organisational justice’ (e.g., manageable workloads; work-life balance; ensuring staff is valued and heard; staff autonomy and control of their work) prevent psychosocial risks at the organisational level [121].

Recommendations for financial stress prevention include staff support hotlines providing financial counselling [122] and HCWs’ right to reimbursement if they are diagnosed with Covid-19 through contact at work [123]. Staff motivation and retention may be enhanced through carefully managed risk ‘allowances’ or compensation [124]. Health systems and providers should also ensure casual nurses are equally supported and protected to other staff during pandemics. This should include offering ongoing, permanent, or fixed-term work during the pandemic, paid sick leave and allowance for self-quarantine if necessary [4,72].

The studies highlight that material and psychological resources should be provided across all the stages of the pandemic and during its aftermath in order to prevent post-traumatic symptoms and to enable the processing of difficult experiences among HCWs.

## 4. Discussion

We reviewed research that has been conducted on psychosocial risks to healthcare workers during the Covid-19 pandemic, identifying categories of risks, as well as the related prevention measures. Although each health system has its own particularities, the impact of communicable disease outbreaks on HCWs have certain aspects that are independent of the national context [125]. Therefore, the reviewed evidence is effective in providing a general overview and give directions on preventative measures. The scoping review framework was well suited for the study, as it enabled us to identify types of existing evidence, to survey how research is conducted on this topic, to identifying key characteristics related to PSR in healthcare during the pandemic, and to identifying knowledge gaps [43].

The reviewed evidence shows that the COVID-19 pandemic carries a significant psychosocial burden for HCWs—especially nurses and women—that affects both frontline and non-frontline workers and results in several physical and psychological disorders within different contexts. The main risks relate to three sources [9]: (1) job contents, resulting from redeployment and sustaining moral injury; (2) work organization, such as lack of communication and work overload/ poor work-life balance; and (3) risks related to the societal context, such as increased violence against HCWs and financial insecurity. These findings resonate with research previously conducted into conditions that lead to PSR in general, and in the healthcare sector specifically. The pandemic has intensified the pre-existing psychosocial risk factors as evidenced by the global data on negative health outcomes.

In addition, a prominent source of PSR is the lack of but also the use of PPE. For the latter, the risk relates to a lack of training for the safe use, and the barriers the PPE forms to verbal and non-verbal communication between care providers and patients. Previous studies into healthcare workers’ adherence with infection prevention and control (IPC) for respiratory infectious diseases have highlighted that the main barriers for safe PPE use include constantly changing policies and high workload, that have been characteristics of the Covid-19 pandemic [126].

The review found a combination of measures to mitigate and prevent PSR. Interventions that seem to have a protective effect for HCWs’ well-being involve management engagement and multi-dimensional approaches requiring material and psychological support. The interventions focus largely on acute individual mental health support, and organisational practices. Recommendations that would address the possible longer term health impacts on HCWs are limited, even though the impact of the Covid-19 pandemic can increase the risk for chronic stress, and the fact that HCWs are often reluctant to seek mental health care [127] can exacerbate the “psychological pandemic” [116].

Constantly changing practices and lack of clear information were found to be stressful for HCWs. The changing epidemiological situation and SARS-Cov-2 virus mutations result in changing practices and policies, and in this situation, clear and effective communication within care institutions is crucial [113]. Relatedly, it is also important to engage with HCWs regarding their concerns that might drive Covid-19 vaccine hesitancy or reluctance [128]. Mandatory training for all staff is an important measure to support the adherence of safe PPE use, and it can simultaneously act as a preventative measure to the anxiety related to the use of PPE [80,129].

In addition to communication to and training of workers, their participation in the planning and management of changes in work organization and OSH would be important for PSR identification and prevention of PSR [14,37,38]. This aspect of PSR prevention was not fully addressed in the reviewed materials.

Furthermore, this review highlights that workplace occupational safety and health measures are the primary level of acute psychosocial risk mitigation, and that there are important aspects of PSR prevention that require urgent attention.

Poor working conditions lower HCW retention rate, and health workforce shortages hinder the prospect of healthy work organisation. 

Moral injury that violates the normative expectations of HCWs is often the result of lapses in leadership at policymaking and managerial level (e.g., scarcity of resources) and organisational level (e.g., redeployment) [118]. 

Financial stress has consequences to the individual HCWs and more broadly; studies on long-term care facility workers in the UK during the Covid-19 pandemic found that staff play a key role in transmitting infection to each other and to the residents. The facilities that provided staff sick pay had significantly fewer cases of infection among both residents and staff compared with those that did not, as the workers could adhere with self-isolation rules and not to take multiple jobs without facing loss of income [130].

Stigmatization of and violence against HCWs have been significant risk factors before and during the pandemic, and the preventative and mitigating measures regarding this PSR require strengthening.

Research into health workers’ working conditions during the pandemic is ongoing, and the emerging studies provide future details on PSRs and their health impact. As many Covid-19 patients have pre-existing comorbidities, advancing research into different types of therapeutic procedures can help to relieve the strain on intensive care units and the workers [131]. Longitudinal studies will provide important information; for example, a study that analyzed the change of Spanish nurses’ mental health status over the Covid-19 outbreak, found that changes occurred in three stages: in the early stage their experience was mainly being ambivalent, as they were torn between a sense of professional mission and fear of being infected; the middle stage was characterized by anxiety, depression, somatisation, compulsiveness, fear, and irritation; and in the later stage psychological adaptation began to occur, through feelings of meaningfulness of the work. According to the study, only variables directly related to the Covid-19 outbreak that were predictive factors of change, over time, in depression, anxiety and stress symptoms were ‘the fear to infect others’ and ‘the fear to be infected’ [132]. In the light of our review, this emphasises the importance of understanding the pathways between biological hazards and psychosocial risks, the importance of monitoring the health and wellbeing of HCWs, evaluation of the efficiency of different interventions, and the development of new risk mitigation strategies and measures to match the lived reality of HCWs.

Exploring differences in the sources of PSR during the different waves of the pandemic could also provide important perspective into the employment and effectiveness of preventative measures (e.g., availability of PPE and training on the safe use) and the emerging OSH measures, in particular those that address social and societal risks that health workers are exposed to, and those that enhance worker participation in OSH.

Analysis of the articles in this review found the following knowledge gaps that future research should focus further on:(a)*The long-term impact of the identified psychosocial factors.* Evidence is needed regarding the psychological and physical health cost that HCWs pay while being exposed to the first as well as the second wave of the pandemic. Longitudinal study designs may be particularly helpful for studying the long-term effect of the pandemic on HCWs well-being, especially in the light of the care that has been postponed due to the pandemic and will burden health systems in the years to come.(b)*Psychosocial risks for different occupational groups of healthcare workers.* Evidence should be collected regarding the risk factors that are particular for speech therapists, paramedics, support, catering, cleaning, and administrative staff in healthcare settings as well as for carers and therapists working in community settings. More evidence is also needed regarding the risk factors for HCWs employed in primary care settings, and social care, as well as in settings that were set particularly for the control of the pandemic (e.g., emergency lines workers, and people who administer Covid-19 tests). The emergence of new variants of the virus, such as the Variant of Concern (VOC) B.1.1.7, with substantially increased transmissibility compared to other variants, calls for new studies on the exposure of healthcare workers.(c)*The intersecting inequalities in psychosocial risks for healthcare workers, particularly for female HCWs.* More evidence is needed regarding the actual mechanisms and contexts that increase intersectional vulnerabilities, such as precarious employment, in order to suggest effective prevention and intervention strategies, including: empirical evidence regarding the impact of the pandemic on the physical and psychological well-being of HCWs who identify as ethnic and racial minorities; and evidence on ethnic and/or racial inequalities in mental health outcomes among HCWs.(d)*Finally, an analysis of the combination of workplace and broader societal policy measures that can prevent and mitigate the identified psychosocial risks to healthcare workers is needed to strengthen the preparedness of health systems for future pandemics.* Since the Covid-19 pandemic is the first to impact European health systems in a century, country specific analyses on HCWs’ working and employment conditions in relation to psychosocial risks merit further research.

### Limitations

The quality of evidence included in a scoping review is not formally evaluated as the information is gathered from a wide range of study designs and methods. The studies reviewed are all in English language. The majority of the European studies focus on Italy and France; and there are many studies focusing on the UK, China, USA, India. The majority of the studies with primary data are cross-sectional, so they provide only a snapshot of the situation and capture only a fraction of experiences, as the evidence does not cover the period of the second wave of the pandemic that seems to be evolving during the end of 2020 and the beginning of 2021 and beyond. The review finds most references in secondary-care and tertiary-care, and less so in primary-care and community settings. Data are vastly representative of doctors and nurses while data on administration and support workers in hospitals as well as on carers that work in community settings are scarce.

## 5. Conclusions

This scoping review shows that psychosocial risks are a continuing concern for healthcare workers during the Covid-19 pandemic. The findings underline that further emphasis on the prevention and mitigation of these risks is essential for occupational safety and health, and for sustainable health systems.

This review largely confirms the findings of reviews that were conducted at an earlier stage of the global Covid-19 pandemic in terms of HCWs negative health outcomes. In addition, the findings on PPE as a psychosocial risk, and the impact of the pandemic on non-clinical healthcare workers and community-based healthcare workers further enrich the knowledge base. The results highlight that while the impact of the variety of psychosocial risks manifests at the individual mental and physical health level, majority of the preventative measures go beyond the individual healthcare worker and should be developed and strengthened further. Specifically, measures that address factors that impact work organisation, such as workforce shortages, and the impact of employment conditions on psychosocial risks, such as financial stress, and the broader intersecting inequalities that determine occupational health disparities, such as gender and ethnicity.

## Figures and Tables

**Figure 1 ijerph-18-02453-f001:**
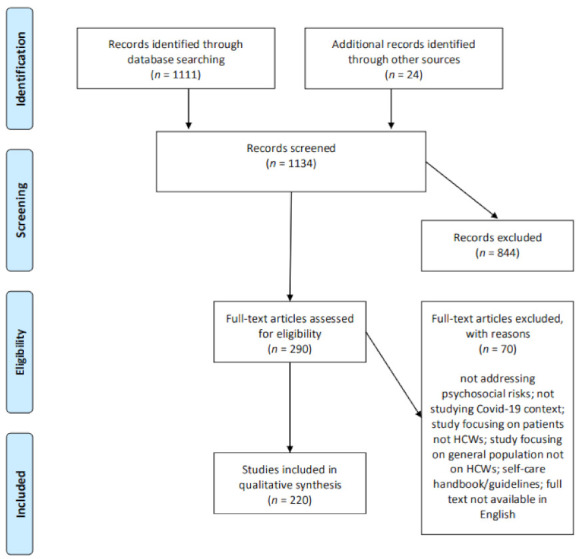
PRISMA flow diagram of the study selection process.

**Figure 2 ijerph-18-02453-f002:**
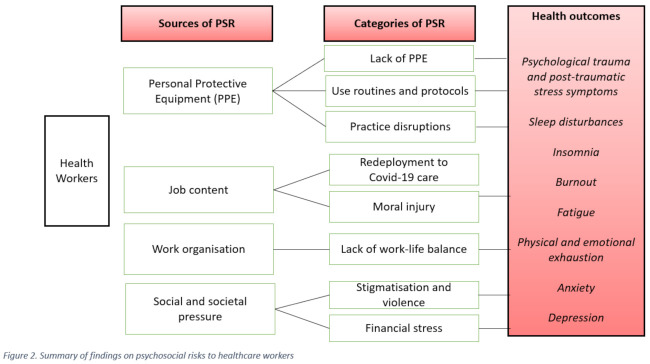
Summary of findings on psychosocial risks to healthcare workers.

**Table 1 ijerph-18-02453-t001:** Settings and occupations studied.

Characteristic	Categories	*N*	(%)
Settings	Hospitals & Medical Centres	101	45.9
	Multiple Settings	14	6.3
	Community Settings	7	3.2
	Primary Care	2	1
	Psychiatry facilities	2	1
	Geriatric facilities	1	0.4
	Testing Clinic	1	0.4
	University	1	0.4
	Not specified	91	41.4

Occupations	Medical HCW	63	28.7
	Nurses	29	13.3
	Doctors	11	5
	Medical & Non-Medical HCW	30	13.8
	Pharmacists	2	1
	Dentists	2	1
	Surgeons	2	1
	Residents	1	0.4
	Paramedics	1	0.4
	Students	1	0.4
	Caregivers	1	0.4
	Not specified	76	34.6

This table includes the numbers and descriptions of the setting and the occupations in the reviewed articles.

## Data Availability

Data is contained within the Appendix A.

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
