# Peer review of "A Scoping Review of Psychosocial Risks to Health Workers during the Covid-19 Pandemic"

_ijerph, 2021, doi:10.3390/ijerph18052453_

Round 1

Reviewer 1 Report

Thank you for the opportunity to review the manuscript ijerph-1110737.

The authors submitted  a Review of psychosocial risks to health workers during the Covid-19 pandemic.

The subject of the is important however major parts of the study must be carefully revised.

Introduction:

Too short to be connected with the results and conclusions. a substantial revision is necessary, alternatively it would be possible to add a background.

Materials and Methods:

  1. 70-72: please use Medical Subject Headings

Results:

Please insert a table with a list of abbreviations.

Integrate studies that address the professional risk of swallowing therapists and speech therapists.

 This is one of the occupational groups that are most at risk of all. In this context, please draw your attention to the deficits of existing research tasks

  1. 314 note the distances from the bullet points.

Figure 2 is difficult to read (dark color). Please integrate in this Figure „work organisation“ the point

„disregard of regulations by the employee himself“.

Discussion:

There is no discussion of the results. Please add references and combine them. Refer to the literature from the backgroud which has yet to be incorporated.

Add also suggested solutions or suggestions for the improvement of structure workflows. For example to protect employees and patients, add reference https://doi.org/10.3390/clinpract11010013. (Complex treatment for infection with pathogens requiring isolation).

Please give an update: Variant of Concern (VOC) B.1.1.7. The mutated virus calls for new studies on the exposure of employees. Discuss also the employees who do not adhere to the protection requirements.

Reviewer 2 Report

In total, 220 articles published between 1 January and 26 October 2020 were included into this review, aimed at identifying key categories of psychosocial risks to healthcare workers during the Covid-19 pandemic. The analysis implies that the pandemic relates to three main categories of psychosocial risks in both clinical and non-clinical health workers: Personal Protective Equipment; Work organization; and Social/societal pressure.

The results are mainly based on studies from the European Union, North and South America, Asia and Australia. The review shows extensive psychosocial risks during the pandemic among healthcare workers all around the world and that women are at greater risk than men. The authors summarize their findings in a systematic way and, on the basis of their findings, they suggest a number of preventive and intervention measures on an individual as well as an organization levels. The authors also identify knowledge gaps that future research should focus on, such as long-term psychosocial risks among healthcare workers and the impact on different occupational groups.

This is an impressive study providing strong evidence for the psychosocial risks among healthcare workers during the Covid-19 pandemic. However, only in a few cases the results are compared to psychosocial risks among healthcare workers during non-pandemic conditions. Therefore, I have the following comment and question.

The review was based on studies published between 1 January and 26 October 2020. This means that most studies were performed before (January 2020), during (March to June, 2020) and after (July and August, 2020) the peak of the first wave of the Covid-19 pandemic. Depending on the level of outbreak of the disease, the pressure on the healthcare workers is assumed to vary considerably between the different studies. Would it be possible to relate the findings from this review to the different phases of the pandemic? For example, the spread of the disease was generally very low during the summer of 2020 and, presumably, the psychosocial risks among these workers would be lower during the summer compared to other periods. If such a relationship could be established, this would indicate the magnitude of the psychosocial risks associated specifically with the peaks of Covid-19 pandemic.

Reviewer 3 Report

Dear authors,

Thank you for the opportunity to read your paper, which I found quite interesting and relevant in our current days.

Although I think the paper is interesting, some issues must be addressed:

1) In the Abstract you should clearly state the knowledge gap your paper aims to fulfill.

2) The aim of the paper that is presented in the Abstract and in the Methods section should be the same. The aim of the paper which is presented in the Methods section is not totally clear, probably because it is too long. Perhaps it would be interesting to divide it into specific objectives.

3) You make reference to several studies in the Introduction section. However, it is too short and, as a consequence, the rationale of your paper is not strength enough. You should present the state of the art on the topic, the knowledge gap you identify, the relevance of fulfilling that gap, and finally the aim of your study (in the "Introduction" section).

4) In the "Methods" section you state that "This scoping review employed the method outlined by Sucharew & Macalus". However, Sucharew and Macalus do not present any method at all. Indeed, what they state in their paper is that "The methodological framework for scoping reviews was developed by Arksey and O’Malley and further refined by Levac et al. and the Joanna Briggs Institute"...

5) You should clearly present your search strategy, in line with the recommendations of the Joanna Briggs Institute, for at least one database. Otherwise, your search cannot be replicated.

6) You should justify all your methodological options: why including papers only in English? Why including papers only from 1st January on? Why not assessing the methodological quality of the papers? Why including grey literature?

7) Why did you decide to search only on PubMed and Google Scholar? Indeed, none of them are databases, but search engines! I think that is one of the most significant limitations of your study.

8) You should provide, as a supplementary file, the whole data chart information. 

9) How did you organize the information in categories? What were the methods used to do that?

10) In your scoping review you included, mostly, cross-sectional studies. However, cross-sectional studies have several limitations, and that is why it is crucial to carry out longitudinal studies. Some have already been conducted in this topic, and I think in the "Discussion" section you should compare your findings with the one of some of those studies (example: https://doi.org/10.1016/j.envres.2020.110620). Moreover, I think you should point out the need to carry out more longitudinal studies

11) The limitations should be presented at the end of the "Discussion" section.

Round 2

Reviewer 1 Report

The authors have implemented the suggestions to improve the article.